# Getting over the hurdles to save lives: Incorporating perceived barriers into theory of planned behaviour (TPB) model to predict stated intention among Hong Kong trained laymen

Victor C.W. Tam[1]*, Nelson C.Y. Yeung[1], Anthony Wai Leung Kwok[2,3]*

**1** Centre for Health Behaviours Research, The Jockey Club School of Public Health and Primary Care, Faculty of Medicine, The Chinese University of Hong Kong, Hong Kong, SAR, China, **2** School of Medical and Health Science, Tung Wah College, Hong Kong, SAR, China, **3** Department of Orthopaedics and Traumatology, Faculty of Medicine, The Chinese University of Hong Kong, Hong Kong, SAR, China

* victorcw.tam@cuhk.edu.hk (VCWT); anthonykwok@twc.edu.hk (AWLK)

## Abstract

### Background

The 'Intention-focused model' is advocated for educating bystanders in Basic Life Support (BLS), consisting of cardiopulmonary resuscitation and automated external defibrillation. Although previous International Consensus on Cardiopulmonary Resuscitation statements have summarised barriers at the scene of out-of-hospital cardiac arrest, few studies have investigated how barriers perceived by trained laypeople affect their intention to deliver BLS (BLS intention) and the underlying mechanisms, especially during the COVID-19 pandemic when global fear of infection was persistently heightened. This study examines the relationship between perceived barriers (PB) and the Theory of Planned Behaviour (TPB) constructs, i.e., attitude, subjective norms (SN), perceived behavioural control (PBC) and BLS intention, as well as the role of concern about infection.

### Methods

A cross-sectional online survey of 678 trained adult laypeople was conducted in 2022 using convenience sampling. Structural equation modelling was used to analyze the relationship between PB subscales, BLS intention, attitude, SN and PBC. Multi-group analysis was used to examine the potential moderating effect of concern about infection.

### Results

A split-construct model consisting of two barrier subscales showed better model fit ($\chi2/df = 3.224$, CFI = 0.966, RMSEA = 0.057) than the lumped-construct model ($\chi2/df = 3.732$, CFI = 0.953, RMSEA = 0.064). Performance-related barriers had a

**Data availability statement:** All relevant data are within the manuscript and its Supporting Information files.

**Funding:** The author(s) received no specific funding for this work.

**Competing interests:** The authors have declared that no competing interests exist.

strong significant association with PBC (β = −0.74, P < 0.001), while cultural barriers were significantly associated with attitude (β = −0.34, P < 0.001), and SN (β = −0.16, P < 0.001) were significant, but not for PBC (β = 0.01, P = 0.905). Attitude, SN, and PBC had significant effects on BLS intention (β = 0.14–0.56, Ps < 0.001). Multigroup analysis showed that concern about infection significantly moderated the relationship between cultural barriers and attitude (Low concern: β = −0.474, P < 0.001; High concern: β = −0.326, P < 0.001; Δχ²(1)=6.319, P = 0.012).

## Conclusions

This study offers empirical evidence for integrating perceived barriers into an intention-focused model to predict laypersons' willingness to perform resuscitation. Future research should focus on addressing priority concerns and transforming them into heightened willingness through enhancing positive attitudes, SN and PBC.

## Introduction

The global overall survival rate of out-of-hospital cardiac arrest (OHCA) is as low as around 10% [1]. In Hong Kong (HK), among 10000 OHCA victims in 2022, only 8.3% of victims survived upon arrival at the hospital emergency unit [2]. Nearly half of the victims (44.6%) received bystander cardiopulmonary resuscitation (CPR). However, very few (1.9%) received bystander defibrillation, despite the number of publicly accessible automated external defibrillators (AEDs) having been increased to around 3500 [3] and enhanced promotion by the government [4].

Incorporating the Theory of Planned Behaviour (TPB), the Intention-focused model is advocated for bystander cardiopulmonary resuscitation and automated external defibrillation, collectively referred to as Basic Life Support (BLS) [5]. According to the model, the three TPB determinants, i.e., attitude, subjective norms (SN), and perceived behavioural control (PBC) are the key constructs to predict BLS intentions. Previous studies have examined this model in undergraduate students [6] and lay-responders [7] in the USA; as well as university students [8] in mainland China, with the variable accounting for 51–57% of the variance in the willingness to provide resuscitation.

Further to the three TPB determinants, common perceived barriers among bystanders have been identified and might affect their intention to deliver BLS (BLS intention). Psychological barriers (e.g., overwhelming fear and distress), practical barriers (e.g., inability to position the patient on a flat firm surface), and cultural barriers (e.g., perceived inappropriateness to closely contact with female patients) to CPR and using AED were evident [8–12]. A local telephone survey with a random sample of 1001 community members also revealed similar barriers to bystander AED in HK [13]. The top three barriers were concerns about harming the victim and/or rescuer (57.3%), perceived lower effectiveness of AED conducted by lay people than healthcare professionals (44.6%), and potential legal liability (42.4%). While limited studies have examined the association between perceived barriers and intention, Chen et.

al. [14] revealed a significant association between perceived barriers and intention (path coefficient = −0.109, p < 0.001), in addition to TPB determinants, among civil servants in mainland China using structured equation modelling. Yet, they only investigate the direct effect of perceived barriers on intention, not the association between perceived barriers, attitude, subjective norms, and PBC, where a qualitative study suggested a potential linkage among these concepts [15]. Furthermore, it is unclear whether barriers across different aspects should be considered as a whole or separately, as they may affect distinct perceptions.

Among all barriers, concerns about contracting infectious diseases have consistently been identified as a perceived barrier among laypeople affecting their intention to perform BLS [11,12], particularly the risk of respiratory infections from mouth-to-mouth ventilation [16]. While the COVID-19 pandemic has struck HK and the rest of the world, it has heightened community members' concerns about infection during potential close contact with others. Yet, a local study showed that levels of fear about infection could vary among individuals [17]. Under these unusual circumstances, limited studies have examined how varying levels of concern about infection affect laypeople perceptions and intentions to perform BLS. Therefore, the aim of this study is twofold. First, we aimed to examine the association between BLS intention, the three TPB determinants, and perceived barriers. Also, this study aimed to examine the moderating effect of concern about infection in this relationship under the COVID-19 pandemic.

## Methods

### Study design and data collection

This article presents a secondary analysis of an online cross-sectional survey conducted in HK from December 28, 2021, to April 19, 2022, using convenience sampling. An anonymous survey link has been distributed through social media, including WhatsApp, Instagram, and Facebook. Among 1449 valid responses collected from local Chinese community members, 678 non-healthcare adults who received previous CPR and/or AED training were extracted and analysed. Fig 1 illustrates the flow of extracting responses. Ethics approval was obtained from the Survey and Behavioural Research Ethics Committee of The Chinese University of Hong Kong (reference no: 158−21). Before participating in the survey, participants were required to provide a written agreement of their implied consent upon completing the online survey voluntarily. The study was conducted in accordance with the Declaration of Helsinki.

### Instruments

Several relevant surveys have been identified through a comprehensive literature review of previous studies [18–21]. Upon permission from the authors, we adopted and modified some items to develop the measurement scale, which comprises 20 items related to psychobehavioural constructs. Two experts and eight target respondents provided feedback on the scale's relevance, importance, and clarity. No item was suggested to be excluded. As suggested by the target participants, definitions and graphical illustrations of 'CPR' and 'AED' have been added to align respondents' understanding. On average, participants took around 15 minutes to complete the survey. The finalised version of the scale consisted of 20 items in five sub-scales: (i)overall BLS intention (2 items), (ii)attitude (4 items), (iii)subjective norms (5 items), (iv)PBC (4 items), and (v) perceived barriers (5 items). All items were assessed using 5-point (0–4 points) Likert scales. Demographic characteristics, such as age, gender, and education level, were collected anonymously.

### Statistical analysis

Data were analysed using IBM SPSS Statistics version 26 (SPSS Inc., Chicago, USA). Descriptive statistics were used to provide an overview response. Cronbach's alpha (α) was used to assess the internal consistency reliability of the subscales, for which an α > 0.7 was considered high reliability [22]. Correlations between variables were assessed using Pearson's correlation (r). Exploratory factor analysis was used to examine the internal structure of the perceived barriers

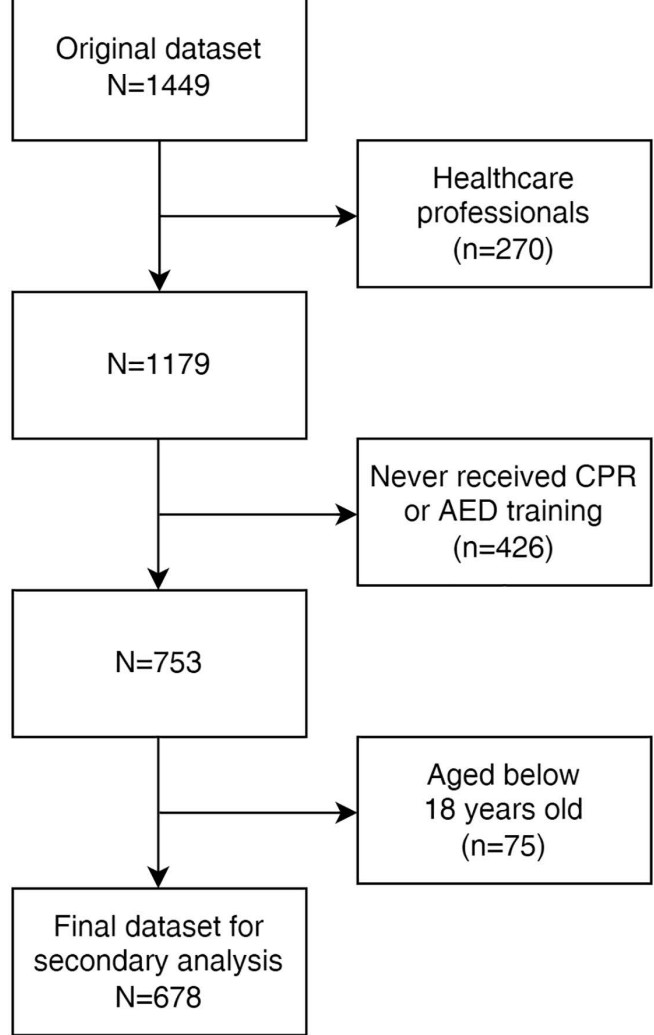

**Fig 1. Steps of responses extraction.**

subscale. Structural equation modelling was conducted using SPSS AMOS to examine the relationship between constructs. Two separate models have been constructed to test the effect of PB as a single construct or as separate barrier subscales. The model fit was determined using the following parameters: the chi-square to degrees of freedom ratio ($\chi^2$/df < 5), comparative fit index (CFI > 0.95), and root-mean-square error of approximation (RMSEA < 0.08) [23].

A multigroup analysis was conducted in SPSS AMOS according to Byrne's standard procedures [24] to examine the moderating effect of concern about infection. Two levels of concern about infection were determined by the response of the item "*I am concerned about getting infections from the victim when performing CPR and using AED*" (Low level = score 0–1; High level = score 2–4). Model invariance was tested using the Chi-Square Difference ($\Delta\chi^2$) test. A model non-invariance was determined by $P < 0.05$. To identify the specific path moderated by the level of concern about infection, the initial screening used the Critical Ratio for Differences (CRD), followed by the $\Delta\chi2$ test for the moderating effect on the target path. A $|Z| > 1.96$ was considered potentially significant moderation in the initial CRD screening, which is then confirmed by a $P < 0.05$ in the subsequent $\Delta\chi2$ test.

## Results

Among 678 non-healthcare participants, 353 (52.1%) were females. Besides, 265 (39.1%) were aged between 18–25 years, 222 (32.7%) were aged between 26–40, and 191 (28.2%) were aged between 41–64 years. For the highest educational level, the majority of them (n = 435; 64.2%) attained post-secondary education, including a diploma or bachelor's degree. For personal monthly income, 255 (37.6%) earned less than HK$15001, 181 (26.7%) earned HK$15001-HK$30000, and 242 (35.7%) earned more than HK$30000.

Perceived barrier revealed, in descending order, were harming the victim (Mean [M]=2.26, standard deviation [SD]=1.16), legal liability if performing BLS wrongly (M = 2.13, SD = 1.23), concerns about infection (M = 1.64, SD = 1.08), time constraint (M = 0.76, SD = 0.78), and getting bad luck from a dying person (M = 0.59, SD = 0.76) (Table 1).

### Reliability

Table 1 shows the descriptives and the reliability statistics of items in each subscale. The Cronbach's alpha for the 'BLS Intention', 'Attitude', 'Subjective norms', and 'Perceived behavioural control' subscales were 0.833, 0.765, 0.916, and 0.937, respectively, thus demonstrating high to excellent internal consistency reliability of the subscales. However, the internal consistency reliability of the 'Perceived barriers' subscale was barely acceptable (α = 0.689).

All participants were invited to participate in the test-retest assessment; 169 (24.9%) completed the retest survey with no missing data. Cohen's kappa ranged from 0.365 to 0.642, indicating fair to moderate reliability of the items. Additional psychometric indicators are presented in S1 and S2 Tables.

### Exploration of barriers subscales

Given the fair internal consistency of the 'Perceived barriers', correlations among individual barrier items were examined to explore possible parcels. 'Harming the victim' and 'legal liability if performing BLS wrongly' demonstrated a strong correlation ($r$ = .62, $P$ < 0.001), thus considered jointly as 'performance-related barriers'. 'Time constraint' and 'getting bad luck from a dying person' also demonstrated a strong correlation ($r$ = .55, $P$ < 0.001), collectively as 'cultural barriers'. 'Concerns about infection' showed weak correlations with the other four items ($r$ ranged .24 to .28).

Exploratory factor analysis revealed two factors with perceived barriers, explaining a total of 67.7% of the variance. 'Time constraint' (factor loading = 0.83) and 'getting bad luck from a dying person' (factor loading = 0.87) were loaded to factor 1, named as 'cultural barriers'. 'Harming the victim' (factor loading = 0.89) and 'legal liability if performing BLS wrongly' (factor loading = 0.87) were loaded to factor 2, named as 'performance-related barriers'. 'Concerns about infection' loaded to both factor 1 (factor loading = 0.47) and factor 2 (factor loading = 0.32). Due to double-loading of the infection item, it was not included in either factor 1 or 2. A separate analysis was conducted to examine the potential moderating role of concerns about infection.

### Correlation

The correlation analysis results are summarised in Table 2. Attitude, subjective norms and PBC were positively associated with BLS intention. Perceived barriers, as a single construct, was negatively correlated with BLS intention ($r$ = −.40, $P$ < 0.001), attitude ($r$ = −.35, $P$ < 0.001), subjective norms ($r$ = −.38, $P$ < 0.001), and PBC ($r$ = −.52, $P$ < 0.001). For the subscales, 'performance-related barriers' demonstrated strong negative correlation with BLS intention ($r$ = −.47, $P$ < 0.001) and PBC ($r$ = −.62, $P$ < 0.001). 'Cultural barriers' demonstrated a weaker negative correlation with BLS intention ($r$ = −.17, $P$ < 0.001), yet a consistent correlation with attitude ($r$ = −.32, $P$ < 0.001). 'Concerns about infection' demonstrated weak but significant correlation with BLS intention ($r$ = −.12, $P$ = 0.003), attitude ($r$ = −.11, $P$ = 0.006), SN ($r$ = −.09, $P$ = 0.018) and PBC ($r$ = −.11, $P$ = 0.006).

**Table 1.** Training backgrounds, knowledge, TPB determinants and barriers (N = 678).

| Continuous predictors | Mean ± SD | Cronbach's alpha |
|---|---|---|
| **BLS Intention** | **2.47 ± 1.09** | **0.833** |
| I am willing to perform CPR on an OHCA victim | 2.45 ± 1.13 | |
| I am willing to use an AED on an OHCA victim | 2.49 ± 1.22 | |
| **Attitude** | **3.44 ± 0.54** | **0.765** |
| CPR and AED are important in saving lives. | 3.69 ± 0.51 | |
| CPR and AED are beneficial to an unresponsive victim. | 3.47 ± 0.72 | |
| It is correct for a community member to perform CPR and use AED. | 3.02 ± 0.91 | |
| It is important for an AED to be available in the public. | 3.60 ± 0.61 | |
| **Subjective norms** | **2.66 ± 0.82** | **0.916** |
| I believe that my family expects me to perform CPR and use AED on an unresponsive victim. | 2.65 ± 0.99 | |
| I believe that my friend … | 2.83 ± 0.96 | |
| I believe that my colleagues/classmates … | 2.87 ± 0.96 | |
| I believe that the local government … | 2.77 ± 0.93 | |
| I believe that the general public … | 2.69 ± 0.91 | |
| **Perceived behavioural control** | **2.44 ± 1.04** | **0.937** |
| Performing CPR and using AED are easy tasks for me. | 2.43 ± 1.10 | |
| I have adequate knowledge and skill to perform CPR and use AED. | 2.56 ± 1.11 | |
| I am confident to perform CPR and use AED. | 2.42 ± 1.16 | |
| I will not hesitate to perform CPR and use an AED on an unresponsive victim. | 2.33 ± 1.17 | |
| **Perceived barriers** | **1.47 ± 0.68** | **0.689** |
| I am concerned about injuring the victim when performing CPR and using AED. | 2.26 ± 1.16 | |
| I am concerned about being sued if I perform CPR and use AED wrongly. | 2.13 ± 1.23 | |
| I am concerned about getting infections from the victim when performing CPR and using AED. | 1.64 ± 1.08 | |
| I am concerned about getting bad fortune when performing CPR and using AED. | 0.59 ± 0.76 | |
| Performing CPR and using AED is too time-consuming. | 0.76 ± 0.78 | |

AED, automated external defibrillator; CPR, cardiopulmonary resuscitation; SD, standard deviation.

## SEM

Two models have been tested. Model 1 represents the lumped construct model, which considers PB as a single construct. Model 2 represents the split-construct model, consisting of two barrier subscales: performance-related barriers and cultural barriers. Model fit statistics indicated good fit for Model 2 ($\chi^2$/df = 3.224, CFI = 0.966, RMSEA = 0.057), which was superior to Model 1 ($\chi$2/df = 3.732, CFI = 0.953, RMSEA = 0.064).

In Model 2, the direct paths from performance-related barriers on attitude ($\beta$ = −0.21, $P < 0.001$), SN ($\beta$ = −0.39, $P < 0.001$) and PBC ($\beta$ = −0.74, $P < 0.001$) were significant. The direct paths from cultural barriers on attitude ($\beta$ = −0.34, $P < 0.001$) and SN ($\beta$ = −0.16, $P < 0.001$) were significant, but not for PBC ($\beta$ = 0.01, $P$ = 0.905). As hypothesised by the TPB, direct paths from attitude ($\beta$ = 0.14, $P < 0.001$), SN ($\beta$ = 0.21, $P < 0.001$), and PBC ($\beta$ = 0.56, $P < 0.001$) on BLS intention were also significant. Fig 2 illustrates the relationship between barriers, TPB determinants and the BLS intention.

**Table 2. Descriptive statistics and correlates of BLS intention and predictors among trained lay-rescuers (N = 678).**

| | 1 | 2 | 3 | 4 | 5 | 6 | 7 | 8 | 9 | 10 | 11 | 12 |
|---|---|---|---|---|---|---|---|---|---|---|---|---|
| 1. Gender | – | | | | | | | | | | | |
| 2. Age | .15** | – | | | | | | | | | | |
| 3. Educational level | −.06 | .04 | – | | | | | | | | | |
| 4. Monthly income | .11** | .56** | .34** | – | | | | | | | | |
| 5. BLS Intention† | .22** | .12** | .003 | .09* | – | | | | | | | |
| 6. Attitude† | .07 | −.04 | .05 | −.06 | .48** | – | | | | | | |
| 7. Subjective norms† | .20** | .04 | −.01 | −.004 | .57** | .50** | – | | | | | |
| 8. Perceived behavioural control† | .28** | .15** | −.07 | .04 | .67** | .45** | .63** | – | | | | |
| 9. Perceived barriers† | −.19** | −.17** | −.03 | −.10* | −.40** | −.35** | −.38** | −.52** | – | | | |
| 10. Performance-related barriers† | −.24** | −.29** | .03 | −.15** | −.47** | −.29** | −.41** | −.62** | .84** | – | | |
| 11. Concerns about infection† | −.08* | .05 | .00 | .07 | −.12** | −.11** | −.09* | −.11** | .61** | .28** | – | |
| 12. Cultural barriers† | −.04 | −.01 | −.12** | −.05 | −.17** | −.32** | −.24** | −.23** | .69** | .31** | .30** | – |

† Highest score = 4.

* *P* < .05.

** *P* < .01.

*BLS*, Basic Life Support.

For the moderating effect of concern about infection, multigroup analysis demonstrated structural non-invariance between the low and high level groups ($\Delta\chi^2$(41)=119.407, *P*<0.001) after confirming a metric invariance ($\Delta\chi^2$(13)=10.127, *P*=0.684). Only the path from cultural barriers to attitude demonstrated moderation in the initial CRD screening (Z=2.448) (Table 3). The subsequent test demonstrated a moderation effect of concern about infection on this path ($\Delta\chi^2$(1)=6.319, *P*=0.012). The relationship between cultural barriers and attitude was stronger at lower levels of concern about infection (Low: *β*=−0.474, *P*<0.001; High: *β*=−0.326, *P*<0.001).

## Discussion

To the best of our knowledge, this was the first local study to investigate the relationship between lay rescuers' BLS intention and perceived barriers. The internal structure of the perceived barriers scale has been explored, including performance-related and cultural barriers. These barrier constructs were associated with BLS intention through the mediation of different TPB determinants. In addition, concern of infection served as a moderator of the relationship between cultural barriers and attitude.

Based on the AHA's 'intention-focused model for bystander', previous studies have shown that all three TPB determinants, namely attitude, subjective norms and PBC, were significant predictors of the intention to perform bystander CPR among undergraduate students [6] and lay-responders [7] in the USA. Among the Chinese population, Xia et al. [25] revealed that knowledge, attitude and subjective norms were significantly associated with intention. In HK, we have demonstrated similar patterns among local community members, with attitude, subjective norms and PBC positively associated with the intention to perform CPR and apply an AED [26]. A previous study has validated a similar TPB-based survey among civil servants in Chongqing, China, with a good model fit ($\chi2$ = 110.809, $\chi2$/df = 1.385, TLI = 0.996, CFI = 0.997, RMSEA = 0.018, SRMR = 0.044) [14]. In addition, this study also demonstrated that perceived barriers were significantly associated with intention (path coefficient = −0.109, p<0.001), which aligned with our findings and supports the addition of perceived barriers scales to enhance current survey tools in measuring CPR willingness using the TPB framework.

While directly addressing the three TPB determinants in the bystander training could be challenging as these constructs tend to be abstract concepts and difficult to be tackled vividly [27], both international and local guidelines advocate

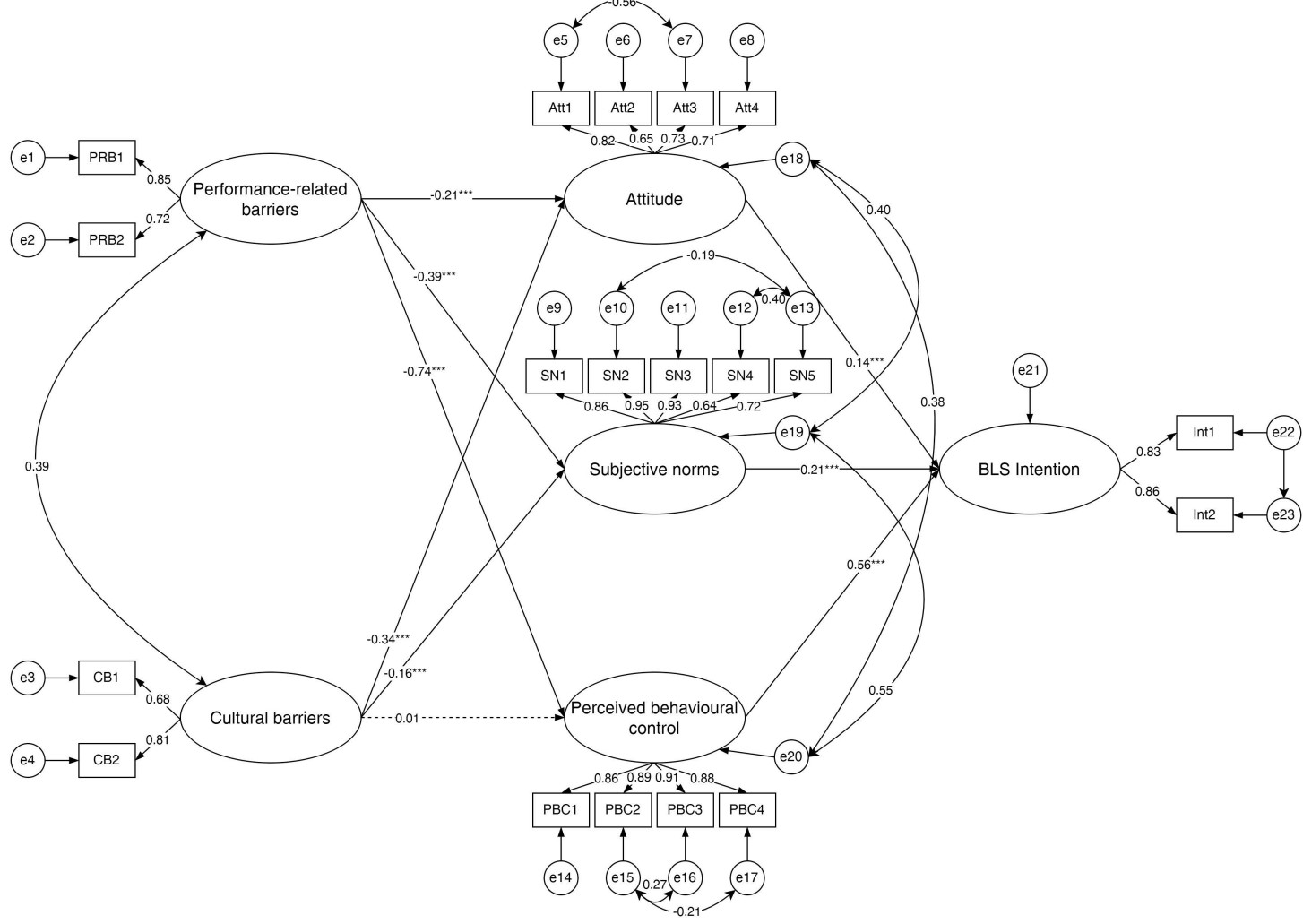

**Fig 2. Structural equation model diagram of the influence of barriers on BLS intention (N = 678).** Standardised regression coefficients were presented. Significant and non-significant structural paths were represented by solid lines and dashed lines, respectively. *P < .05, **P < .01, ***P < .001.

for shifting the primary focus of laypeople's concerns and barriers to start resuscitation [5,28]. However, limited studies investigated the interplay between perceived barriers, attitude, subjective norms, PBC and intention. Therefore, it is unclear whether addressing perceived barriers alone or incorporating TPB determinants could better enhance resuscitation intention. Our findings revealed that reduced perceived barriers were associated with heightened willingness, mediated by the three TPB determinants. This provides an imperative insight into developing an intention-focused pedagogy, not only by addressing bystanders' concerns and barriers but also by emphasising how reduced perceived barriers can be transformed into greater willingness through positive changes in attitude, subjective norms, and PBC.

In addition, our findings revealed that distinct barriers were associated differently with BLS intention through TPB determinants. For instance, performance-related barriers were more strongly associated with PBC than with attitude and SN. In contrast, cultural barriers were strongly associated with attitudes, but were not significantly associated with PBC. While limited studies have examined the association between perceived barriers and the TPB determinants in the context of bystander resuscitation, a recent qualitative study in China [29] delved into lay rescuers' perceptions of bystander

**Table 3. Critical Ratio for Differences for screening of moderated paths.**

| Paths | Z-score |
|---|---|
| Attitude ◊ BLS Intention | −0.424 |
| Subjective norms ◊ BLS Intention | 0.364 |
| Perceived behavioural control ◊ BLS Intention | −0.704 |
| Performance-related barriers ◊ Attitude | 1.451 |
| Performance-related barriers ◊ Subjective norms | 1.686 |
| Performance-related barriers ◊ Perceived behavioural control | 0.847 |
| Cultural barriers ◊ Attitude | **2.444** |
| Cultural barriers ◊ Subjective norms | 1.670 |
| Cultural barriers ◊ Perceived behavioural control | 1.671 |

Note. Potentially significant moderation by the level of concerns about infection was indicated by $|Z| > 1.96$.

resuscitation using the TPB framework. Echoing our findings, the culturally rooted fear of death, which is intrinsically bound with CPR, has emerged as a theme regarding attitude. Also, perceived low skill efficacy and fear of involvement in legal disputes have been identified as main themes under the PBC. The emerging evidence has provided insights and direction for future training strategies for lay rescuers, targeting specific psychobehavioural constructs to promote the translation of lowered perceived barriers into an enhanced willingness to perform resuscitation. For instance, the government could initiate public education and promotion campaigns to address cultural barriers by targeting deep-rooted beliefs and myths (attitude) and fostering a communal sense of helping OHCA victims (subjective norms) towards bystander resuscitation. To address performance-related barriers, in addition to reinforcing self-efficacy through post-dispatcher guidance and advanced simulations in training, strengthening legal protection for bystanders and promoting the spirit of the 'Good Samaritan', even if the resuscitation was not perfectly executed, would be helpful.

While concern about infection was consistently identified as an important barrier in the literature, it appeared as a distinct factor from other barriers in our study. This may be due to the fact that the data collection was conducted during the COVID-19 pandemic, when the fear of infection was unusually heightened. Echoing the International Liaison Committee on Resuscitation (ILCOR) review in 2024 [11], 14 out of the 23 new studies reported that the COVID-19 pandemic was a significant factor in reducing bystander CPR and defibrillation attempts. In addition, our studies revealed that concerns about infection moderated the relationship between cultural barriers and attitudes, with a stronger relationship at lower levels of concern. This suggested that huge concerns about infection weakened the effect of cultural barriers on attitudes, with the fear of contracting infectious diseases likely to outweigh cultural factors during the pandemic. To promote bystander resuscitation, aiming to reduce the fear of contracting infectious diseases perceived by lay rescuers, particularly during rescue ventilation, compression-only CPR, also known as hands-only CPR, has been advocated since 2008 [16]. It has also been reinforced by international resuscitation societies during the COVID-19 pandemic [30].

Despite the AHA having since advocated the 'Intention-focused model for bystanders' in 2022 [5], there are still no comprehensive guidelines or training materials that integrate this framework into standard HO-CPR and AED training modules. Most current bystander resuscitation training courses remain focused on skill and knowledge acquisition, with limited attention to the psychobehavioral factors that influence resuscitation intention. Recently, ILCOR has called for urgent action to address laypeople's concerns and barriers to promote bystander resuscitation rates [11]. Enormous efforts should be directed to develop 'intention-focused' training modules to enhance or supplement current training pedagogies, emphasising the translation of reduced barriers to increased intention by enhancing positive attitude, SN and PBC.

For instance, audio-visual modules can be used to achieve this goal. A recent RCT conducted in Iran found that an educational module comprising videos and PowerPoint slides increased willingness to perform CPR ($p < 0.001$) among

high school students [31]. An ultra-short video instructed module, as brief as 60 seconds, could impact bystanders' skills and perceptions of CPR [32]. These findings demonstrated the possibilities of utilising technologies in achieving 'intention-focused' teaching. Coupling with case-based simulation, situational concerns and culturally salient barriers could be discussed during the debriefing session, facilitated by the instructor [33]. These foster lay rescuers' reflections on their potential choices of action when they encounter OHCA in the future.

## Limitations

There were several limitations in this study. First, the study employed convenience sampling via social media platforms, which is susceptible to sampling biases. Volunteer bias is plausible as participants with greater interest or confidence were more likely to respond. Unknown response rate fails provide an estimation on the level of non-response bias, limiting assessment of representativeness. Digital-only sampling may also have excluded lay rescuers with limited access to or engagement with online platforms, further threatening the sample representativeness. Second, a few items were included in each barrier construct, which may have compromised internal consistency and validity. Additionally, this scale only includes a few commonly mentioned barriers from previous literature, which may not capture the priority barriers among locally trained lay rescuers. Future studies may incorporate additional culturally salient barriers by conducting in-depth interviews with lay rescuers in Hong Kong.

Regarding the SEM analysis, potential construct overlapping of highly correlated constructs, e.g., PBC and Performance-related barriers, may challenge the assumption of unique contributions of constructs and the validity of theoretical distinctions. However, it may highlight the need for future research to disentangle these dimensions more clearly and to determine how we should address these constructs to enhance BLS willingness among trained laypeople. Given concerns about model stability and the lack of validation across samples when targeting lay people, future validation studies are warranted to further confirm the current findings regarding the intention-focused model incorporating perceived barriers. Finally, although TPB-based pathways imply directional associations, the cross-sectional design precludes causal inference, and the observed relationships should therefore be interpreted cautiously and not misconstrued as indicative of temporal or causal effects.

## Future directions

Future research should focus on understanding the interplay of perceived barriers and other psychobehavioural constructs. Development of innovative interventions to address psychosocial barriers and translate into enhanced willingness to perform BLS is warranted. A robustly designed clinical trial to evaluate the effectiveness of incorporating these 'intention-focused' interventions into current lay rescuer training is imperative.

## Conclusions

To conclude, this study provided empirical data to incorporate perceived barriers into the 'intention-focused model for bystanders' to address lay rescuers' resuscitation intention. Barriers could be further divided into performance-related and cultural barriers, each associated differently with intention through TPB constructs. Concerns about infection emerged as a distinct factor during the COVID-19 pandemic. Future research and interventions to enhance bystanders' intention to perform resuscitation should focus on addressing their priority concerns and transforming these concerns into a heightened willingness by enhancing positive attitudes, subjective norms, and PBC. The localisation of 'intention-focused' pedagogy warrants greater attention due to fundamental cultural differences across populations and regions.

## Supporting information

**S1 Table. Additional internal consistency indicators (N = 678).**
(DOCX)

**S2 Table. Heterotrait-Monotrait (HTMT) ratios (N = 678).**
(DOCX)

**S3 Table. Minimal anonymized dataset.**
(XLSX)

## Author contributions

**Conceptualization:** Victor C.W. Tam, Nelson C.Y. Yeung, Anthony Wai Leung Kwok.

**Data curation:** Victor C.W. Tam.

**Formal analysis:** Victor C.W. Tam, Nelson C.Y. Yeung.

**Investigation:** Victor C.W. Tam.

**Methodology:** Victor C.W. Tam, Nelson C.Y. Yeung, Anthony Wai Leung Kwok.

**Project administration:** Victor C.W. Tam.

**Resources:** Anthony Wai Leung Kwok.

**Supervision:** Anthony Wai Leung Kwok.

**Validation:** Anthony Wai Leung Kwok.

**Visualization:** Victor C.W. Tam.

**Writing – original draft:** Victor C.W. Tam.

**Writing – review & editing:** Victor C.W. Tam, Nelson C.Y. Yeung, Anthony Wai Leung Kwok.

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
