## [Decision Letter · Decision Letter 0]

9 Mar 2026

PONE-D-25-57612Validation Of The Intention To Deliver Basic Life Support Scale (I-DO-BLS Scale): Incorporating Perceived Barriers Into Theory Of Planned Behaviour (TPB)-Based Survey To Measure Stated Intention Among Hong Kong Trained LaymenPLOS One

Dear Dr. Tam,

Thank you for submitting your manuscript to PLOS ONE. After careful consideration, we feel that it has merit but does not fully meet PLOS ONE’s publication criteria as it currently stands. Therefore, we invite you to submit a revised version of the manuscript that addresses the points raised during the review process. ======================================This manuscript addresses an important and timely topic and demonstrates adequate theoretical grounding. The sample size is sufficient, and SEM has been employed. However, there are notable technical concerns that require clarification prior to publication consideration. In particular, several standardized path coefficients exceed ±1, which raises concerns regarding model specification, multicollinearity, or reporting accuracy. Additionally, inconsistencies between bivariate correlations and structural estimates remain unexplained. Construct validity reporting is incomplete, and the limitations section does not sufficiently address methodological constraints.

While the manuscript has potential, substantial revision and clarification of the structural model are necessary before the findings can be interpreted reliably.

We look forward to receiving your revised manuscript.

Kind regards,

Nik Hisamuddin Nik Ab. Rahman

Academic Editor

PLOS One

Journal Requirements:

2. We noted in your submission details that a portion of your manuscript may have been presented or published elsewhere.

This work presents a secondary analysis of the previous work titled ‘Evaluation of the

willingness to perform cardiopulmonary resuscitation (CPR) with automated external defibrillator (AED) among Hong

Kong Chinese using the theory of planned behaviour framework: an online cross-sectional survey’. No analysis has

overlapped with the previous work.

4. We note that your Data Availability Statement is currently as follows: All relevant data are within the manuscript and its Supporting Information files

Reviewers' comments:

Reviewer's Responses to Questions

**Comments to the Author**

1. Is the manuscript technically sound, and do the data support the conclusions?

Reviewer #1: Yes

Reviewer #2: Partly

2. Has the statistical analysis been performed appropriately and rigorously? 

Reviewer #1: Yes

Reviewer #2: N/A

3. Have the authors made all data underlying the findings in their manuscript fully available?

Reviewer #1: Yes

Reviewer #2: No

4. Is the manuscript presented in an intelligible fashion and written in standard English?

Reviewer #1: Yes

Reviewer #2: Yes

5. Review Comments to the Author

Reviewer #1: 1) Theoretical and Applied Contribution

The manuscript represents a relevant contribution by integrating perceived barriers within the Theory of Planned Behavior (TPB) and empirically demonstrating how these barriers influence the intention to act through the model’s three core determinants: attitude, subjective norms (SN), and perceived behavioral control (PBC).

The strength of the study lies in the fact that it does not simply describe obstacles but proposes an explanatory and applicable model that helps to understand how and why certain barriers reduce the willingness to intervene. Theoretically, it expands the TPB to contexts where the decision to act must be made in seconds (such as in out-of-hospital cardiac arrests), while practically, it translates the findings into specific educational recommendations: reinforcing normative messages, self-efficacy, social modeling, and legal reassurance.

This connection between theory and application makes the study a useful tool for CPR/AED trainers, health organizations, and curriculum designers involved in prevention and emergency response programs.

2) Analytical Rigor

The analysis is consistent with the objectives and demonstrates a solid statistical structure. First, the exploratory factor analysis (EFA) helped to define the internal structure of the perceived barriers. Then, the structural equation modeling (SEM) compared two approaches: a unified (“lumped”) model and a divided (“split”) model with three subfactors (cultural, performance, and infection-related barriers), showing a better fit for the split model.

This indicates that barriers do not operate through the same psychological pathways and supports differentiating them for educational and diagnostic purposes. The reporting of internal consistency (Cronbach’s), correlations, and test–retest adds further evidence of reliability and temporal stability. Although additional details about normality testing or estimators would be helpful, the analysis meets international standards for validation and structural modeling in behavioral sciences.

3) Public Health Relevance

The study identifies three types of barriers, cultural, performance-related, and infection-related, with direct implications for intention-focused education. The findings can be translated as follows:

Cultural barriers: Address deep-rooted beliefs (fear, superstition, gender roles) and strengthen social norms.

Performance barriers: Reinforce self-efficacy, guided practice, and simulations.

Infection barriers: Clarify safety protocols and promote compression-only CPR.

The research goes beyond the local context, as the mechanisms it explains allow for adaptation to other populations facing similar cultural or legal barriers. It is a valuable contribution to public health education and civic engagement in emergency response.

Points for Improvement

1) Sampling Details

The text alternates between convenience and snowball sampling. It would be useful to unify the terminology and clarify the recruitment process (platforms used, filters applied, response rate, and sample characteristics).

Conclusion

In summary, the manuscript is technically sound, theoretically well-grounded, and has clear applied potential. It offers both scientific rigor and practical relevance, positioning itself as a useful reference for developing intention-focused training in bystander resuscitation and other public health interventions.

Reviewer #2: Dear Author,

This manuscript addresses an important public health topic and applies the Theory of Planned Behavior (TPB) framework to examine BLS intention among lay individuals in Hong Kong. The sample size is adequate, and the use of structural equation modeling (SEM) is appropriate in principle. However, several methodological and statistical concerns must be clarified before the findings can be interpreted with confidence.

1. Several reported standardized path coefficients exceed the theoretical bounds of ±1 (e.g., β = -1.41; β = -1.65; β = 1.24). Standardized regression coefficients should fall between -1 and +1. These values raise concerns regarding model misspecification, multicollinearity, suppression effects, or possible reporting errors (standardized vs. unstandardized estimates). Please clarify and provide appropriate diagnostics.

2. Bivariate analyses indicate that concerns about infection are negatively correlated with intention and TPB determinants. However, the SEM results show strong positive structural paths. This discrepancy requires statistical and theoretical explanation.

3. Given the high inter-correlations among constructs (e.g., r = 0.84), Cronbach’s alpha alone is insufficient. Please report additional psychometric indicators such as AVE, Composite Reliability, and discriminant validity metrics (e.g., Fornell-Larcker criterion or HTMT ratios).

4. The study employed convenience sampling via social media platforms. While acknowledged, the implications of self-selection bias, unknown response rate, and digital sampling limitations require deeper discussion.

5. The proposed TPB pathways imply directionality; however, the cross-sectional design precludes causal inference. This limitation should be more explicitly acknowledged.

6. The transition from 1449 respondents to 678 analyzed participants is insufficiently described. Please provide a detailed breakdown (preferably with a flow diagram).

7. The reported monthly income categories (e.g., <$15,000) appear implausible and may reflect a unit or currency misclassification. Please clarify.

8. The statement claiming this is the first study examining BLS intention among “lay rescuers in HK” appears overly broad. The sample consists specifically of self-reported trained lay rescuers recruited via social media. The wording should be revised to avoid overgeneralization.

9.

The current limitations section does not adequately address:

Model instability concerns

Construct overlap

Cross-sectional design constraints

Potential common method bias

A more comprehensive limitations discussion is necessary.

Kind Regards,

6. PLOS authors have the option to publish the peer review history of their article (what does this mean?). If published, this will include your full peer review and any attached files.

Reviewer #1: No

Reviewer #2: No

---

## [Author Response · Author response to Decision Letter 1]

8 Apr 2026

Responses to reviewers have been uploaded. Thank you again for your effort in reviewing our manuscript and all constructive comments.

---

## [Editor Report · Decision Letter 1]

13 May 2026

Getting Over the Hurdles to Save Lives: Incorporating Perceived Barriers Into Theory Of Planned Behaviour (TPB) Model To Predict Stated Intention Among Hong Kong Trained Laymen

PONE-D-25-57612R1

Dear Dr. Tam,

We’re pleased to inform you that your manuscript has been judged scientifically suitable for publication and will be formally accepted for publication once it meets all outstanding technical requirements.

Kind regards,

Nik Hisamuddin Nik Ab. Rahman

Academic Editor

PLOS One
---

## [Editor Report · Acceptance letter]

PONE-D-25-57612R1

PLOS One

Dear Dr. Tam,

I'm pleased to inform you that your manuscript has been deemed suitable for publication in PLOS One. Congratulations! Your manuscript is now being handed over to our production team.

Kind regards,

on behalf of

Professor Dr Nik Hisamuddin Nik Ab. Rahman

Academic Editor

PLOS One